# 3D Screen Printing Offers Unprecedented Anticounterfeiting Strategies for Oral Solid Dosage Forms Feasible for Large Scale Production

**DOI:** 10.3390/pharmaceutics16030368

**Published:** 2024-03-05

**Authors:** Nicolle Schwarz, Marcel Enke, Franka V. Gruschwitz, Daniela Winkler, Susanne Franzmann, Lisa Jescheck, Felix Hanf, Achim Schneeberger

**Affiliations:** Laxxon Medical GmbH, Hans-Knöll-Street 6, 07745 Jena, Germany; nicolle.schwarz@laxxonmedical.com (N.S.); marcel.enke@laxxonmedical.com (M.E.); franka.gruschwitz@laxxonmedical.com (F.V.G.); daniela.winkler@laxxonmedical.com (D.W.); susanne.franzmann@laxxonmedical.com (S.F.); lisa.jescheck@laxxonmedical.com (L.J.); felix.hanf@laxxonmedical.com (F.H.)

**Keywords:** 3D printing, additive manufacturing, anti-counterfeiting, tablet geometry, 3D screen printing, drug delivery system, solid oral dosage forms

## Abstract

A threat to human health in developed and, in particular, in developing countries, counterfeit medicines represent the largest identified fraud market worldwide. 3D screen printing (3DSP), an additive manufacturing technology that enables large-scale production, offers unique opportunities to combat counterfeit drugs. One such possibility is the generation of oral dosage forms with a distinct colored inner structure that becomes visible upon breakage and cannot be copied with conventional manufacturing methods. To illustrate this, we designed tablets containing a blue cross. Owing to paste properties and the limited dimensions of the cross, the production process was chosen to be continuous, involving two screen and paste changes. The two pastes (tablet body, cross) were identical except for the blue color of the latter. This ensured the build-up and mechanical stability of the resulting tablets in a mass production environment. The ensuing tablets were found to be uniform in weight and size and to comply with regulatory requirements for hardness, friability, and disintegration time (immediate release). Moreover, all tablets exhibited the covert anticounterfeit feature. The study delivers a proof-of-concept for incorporating complex structures into tablets using 3DSP and showcases the power of the technology offering new avenues for combating counterfeit drugs.

## 1. Introduction

Since the early 1990s, an increasing number of counterfeit drugs in circulation can be observed [1,2]. Counterfeit medicines amount to an estimated USD 200 billion market annually, making it the largest identified fraud market globally [3]. In 2022 there were 6615 pharmaceutical crime incidents, corresponding to an increase of 10% compared to 2021 [4]. The World Health Organization (WHO) reports that 10% of the global pharmaceutical trade [5] and as much as 25% in developing countries [6] involves counterfeit drugs. Some sources even state that in in parts of Africa and Asia, this figure exceeds 50% [4,7]. In 2021 Interpol’s worldwide operation discovered more than 3 million counterfeit medicines and medical devices in the UK supply chain worth GBP 9 million [8]. Another operation by Europol seized more than 25 million units of medicines with a value of around EUR 63 million [9]. Already 10 years ago, piracy and counterfeiting cost US businesses more than USD 200 billion annually [10] and causes a total loss of life of between 100,000 and 1,000,000 people worldwide per year [11]. Highly prone to counterfeiting are anti-malaria drugs such as artesunate, where 30 to 50% of all packets bought in southeast Asia were fake [12,13]. Thus, the counterfeiting of pharmaceutical products is a global challenge [5]. According to the WHO definition, a counterfeit medicine is “A medicine that is deliberately and fraudulently mislabeled with respect to identity and/or source. Counterfeiting can apply to both branded and generic products and counterfeit products may include products with the correct ingredients or with the wrong ingredients, without active ingredients, with insufficient active ingredients or with fake packaging” [14]. This poses a growing threat to public health as they can deliver hazardous treatment or even cause death [15].

Counterfeit drugs are hard to detect. Possible techniques for detection are visual features such as packaging, labeling, including spelling mistakes and product security features, and physical (surface discoloration, disintegration, dissolution), or chemical analysis (spectroscopy, chromatography, spectrometry) [16,17,18]. This emphasizes the need for the development of advanced technologies for improved anti-counterfeiting measures for drugs, which should be hard to duplicate, easy to prepare, high-throughput, convenient to recognize, inexpensive [19,20], and legally accepted. However, the FDA supports a wide range of anti-counterfeiting techniques and detection methods [14,21].

Several anti-counterfeiting technologies [22] include: tamper-resistant packaging [23,24,25], track and trace technology [26,27], and product authentication. These can either be covert [16] (such as embedded images, digital watermarks) or overt features [15,19,28]. Even features of a forensic nature, where chemical or biological taggants are introduced into a tablet, are already in development [29,30,31,32].

However, the forensic technique requires licensed technologies for read-out and is thus expensive and unlikely to be available to authorities and the public. The currently most employed anti-counterfeiting measures are data carriers, such as RFID (radio-frequency identification) tags [33,34], or 2D identification methods [19,35,36]. For example, Pfizer uses RFID tags for its Viagra^®^ product, GlaxoSmithKline for Trizivir^®^, and Purdue Pharma labels its Oxycodone product with an RFID tag [33]. Several drawbacks of these anti-counterfeiting measures are that they need to be varied frequently, but patients and the authorities would need information to know which features to look for. Furthermore, covert, or overt features on the packaging would be useless if the product is repackaged.

Therefore, the implementation of anti-counterfeiting measures directly onto the tablet is a necessity. If possible, even a combination of hidden anti-counterfeiting measures and easily recognizable visual features would be the goal.

Herein, we present a novel approach to anti-counterfeiting techniques specifically utilizing the advantages of 3D screen printing (Laxxon Medical GmbH, Jena, Germany, US Patent 20140065194). By employing the 3D screen printing technique where an aqueous dispersion of excipients and API are printed through a screen mesh onto a substrate, thousands of strongly defined tablets can be printed per screen simultaneously. The geometry is defined by the layout of an impermeable emulsion which blocks certain areas on the mesh. After one printing cycle, the deposited layer is dried. The next layer is printed precisely on top of the previous one after lifting the screen by thickness of the dried layer. Subsequently, a three-dimensional geometry is constructed [37,38,39].

Laxxon Medical Corp. (New York, NY, USA) already developed a proof-of-concept work with 3DSP (SPID^®^-Technology, Stetten, Switzerland) having shown its anticounterfeiting potential by printing QR codes directly on tablets as small as r = 5 mm during and as integral part of the manufacturing process [39].

Since 3D screen printing with SPID^®^ technology is a controlled and licensed manufacturing technology that is not generally available, it is impossible to mimic these features in an uncontrolled environment and without extensive technological and development resources. Therefore, this technology does not lend itself to easy duplication.

In addition to the proof-of-concept work on overt anti-counterfeiting measures such as QR-codes imprinted to the surface of a tablet [39], 3DSP offers a new level of anti-counterfeiting strategies, that is, the implementation of covert features within oral dosage forms, which are impossible to counterfeit using conventional techniques. In this paper we present a proof-of-concept study for embedding a covert cross structure into a tablet. With only two screen and paste changes, a structure can be integrated into a tablet which exhibits a distinct pattern upon the breakage of the tablet in either the middle or slightly off-center. Additionally, the only difference in the paste compared to the overall tablet material is the addition of food coloring, meaning only a subtle change in API content due to the addition of food coloring and no strongly irregular API distribution in the tablet. This means that no alteration in the biopharmaceutical performance would be expected compared to a homogenous tablet without the embedded cross.

## 2. Materials and Methods

### 2.1. Chemicals

Materials used are listed along with their providers in brackets: Paracetamol (acetaminophen; Caelo (Hilden, Germany), Avicel PH-105 (Dupont, Wilmington, DE, USA), glycerol (AppliChem, Darmstadt, Germany), calcium sulfate (Honeywell, Morristown, NJ, USA), talc (Carlo ERBA, Germany), silfar (Wacker Chemie, Germany), hydroxypropyl cellulose (Klucel^®^ LF, Ashland, Wilmington, DE, USA ) and Polyplasdone Ultra 10 (Ashland, Covington, KY, USA), pharmatose 450 M (DFE Parma, Goch, Germany), blue dispersible color (TopMill^®^ blue 260.36) was kindly provided by Biogrund (Huenstetten, Germany). Milli-Q water (18.2 MΩ cm) was used for all formulations and solutions.

### 2.2. Preparation of Printing Pastes

All pastes (1 and 2, see Table 1 and Table 2) were mixed with a vacuum dissolver planetary mixer HRV-S 2DP from Herbst Maschinenfabrik GmbH (Buxtehude, Germany) with 3 agitator tools in a 2-L glass container.

P1, used to build the tablet body, was generated at room temperature according to the composition in Table 1. Briefly, a 10 wt% gel of Klucel LF Pharma was prepared. The appropriate amount of binder gel (Table 1, Klucel^®^ LF Pharma) was then added into the 2-L glass container of the planetary mixer. In the next step, all other ingredients were added into the glass container. Mixing was performed for 20 min at 1000 rpm under 70 mbar vacuum.

P2 for the blue cross was formulated according to the composition in Table 2 under room temperature conditions. Its preparation followed the path outlined for P1 applying the same mixing conditions.

### 2.3. Rheology Measurements

Rheology measurements were performed on a Kinexus Rheometer (Netzsch, Selb, Germany), equipped with a plate-plate geometry and a passive solvent trap at 25 °C. A 500 µm gap was selected. In the 3-step shear rate test, shear rates of 0.1 s^−1^ (for 1.5 min), 100 s^−1^ (for 0.5 min) and 0.1 s^−1^ (for 10 min), were applied. The amplitude sweep was conducted at 1 Hz, and the frequency sweep at 0.05% strain.

### 2.4. 3D Screen Printing of Tablets

Tablets were fabricated on the prototype inline unit EX301i from Exentis Group AG (Stetten, Switzerland). Their lateral dimensions were determined by the used screen layouts (for example, see Figure 1). Machine preparation for the printing process included mounting and alignment of the screen (Pröll Services GmbH, Hildesheim, Germany) and the squeegees, as well as the set-up of the software SIMPLEX EKRA HMI 2010 (Esp021-104-2021-09-08-0). Printer settings used were as follows: flooding squeegee speed 100 mm s^–1^, printing squeegee speed 65 mm s^–1^, off-contact distance 2 mm, height increment for screen elevation 60 µm, dryer temperature 80 °C (convection dryer), and speed of conveyor belt under convection dryer 2 m min^–1^. Using these settings, 55 layers were printed. To enable a colored cross in the middle of the tablets, the two pastes and the two screens needed to be changed during the printing process. Firstly, Paste 1 was used for 25 layers with Screen 1 (Figure 1), then Paste 2 was utilized to print the colored cross for 5 layers on top with Screen 2 (Figure 1). Afterwards 25 layers of Paste 1 were printed to finish the tablets using Screen 1.

Tablets were printed on tempered glass plates (300 × 300 × 5 mm^3^) with a satin-finished surface. Upon completion of the printing process, tablets were left on the plates for overnight air-drying at room temperature before they could be readily displaced by applying tangential mechanical force with a razor blade.

The inline printer was equipped with 10 glass plates (capacity: n = 13). For demonstration reasons, one printing plate was removed during the printing step after printing the colored cross to visualize the inside geometry of the tablet.

### 2.5. Tablet Hardness and Friability Testing

Resistance to crushing was determined according to Ph. Eur. 2.9.8 (10). Five tablets of each size and shape were tested with a Pharmatest PTB 511E (Pharma Test Apparatebau AG, Hainburg, Germany). Friability was assessed according to Ph. Eur. 2.9.7 (10). Approximately 6.5 g tablets of each size and shape were dedusted with compressed air, weighed, placed in the friability tester Pharmatest PTF 100 (Pharma Test Apparatebau AG, Hainburg, Germany) and rotated 100 times at a constant speed of 25 rpm. Thereafter, they were removed, dedusted and weighed.

### 2.6. Mass Uniformity

The weight variation test was used to assess mass uniformity. The individual weight of 20 tablets from each formulation/geometry was determined, and the average/variation was calculated.

### 2.7. Dimension Uniformity

Dimension uniformity was quantified with the dimension variation test. The 3 dimensions of 20 tablets from each formulation/geometry were measured with a Digimatic Caliper CD-15APX (certified) and the average/variation was calculated. Mean and standard deviations of the surface area SA, volume V, surface-area-to-volume-ratio SA/V and density ρ of the tablets were calculated based on these measured values.

### 2.8. Disintegration Testing

Disintegration was tested according to Ph. Eur. 2.9.1. Six tablets of each size and shape were applied to a Pharmatest Auto1EZ (Pharma Test Apparatebau AG, Hainburg, Germany), an automatic disintegration time tester using discs and demineralized water (750 mL, 37 °C) as medium. Data are reported as average ± standard deviation.

### 2.9. Dissolution Testing

Dissolution experiments were performed according to the USP monograph “Acetaminophen tablets” section dissolution of the US Pharmacopeia. An Agilent^®^ dissolution paddle apparatus equipped with an Agilent sampling station was used. Testing was performed using 900 mL phosphate buffer, pH = 1.2 at 37.0 ± 0.5 °C, with stirring at 50 rpm. Sink conditions were used.

Analysis (n = 6) was performed using an Agilent 1260 Infinity II HPLC, equipped with a Kromasil C-18 250 × 4.6 mm 5 µm column (Nouryon AB, Göteborg, Sweden) at a temperature of 35 °C, 20 µL injection volume and measured at an absorption of λ = 243.

A calibration curve in the range of 6.25–400 µg/mL Paracetamol was made, which correlated with R = 0.9999.

## 3. Results and Discussion

### 3.1. Printing Tablets of Different Geometry and Size

In this study, two screens with different layouts (disk (Screen 1) and cross (Screen 2)) were used (Figure 1, Table 3). The resulting tablets have a disk shape with an integrated cross. The shape of the resulting tablet is defined by the screen layout. Only the areas where the mesh is not covered with emulsion (green area) allow for the extrusion of paste in this region. Therefore, the lateral tablet dimensions (diameter *D*, length *L* or width *W*) are specified by the respective 2D layouts on the printing screen. However, the tablet height is dependent on the number of printed layers and their thickness which was set here to 55 layers and resulted in a tablet heigh of 2.8 mm (Table 3).

The surface of the printing plate occupied by the tablets was about 40% of the printable plate surface. Using this research and development set-up, the overall batch size was 3970 tablets (330 tablets × 9 plates). The entire process (printing and drying) took 3 h and 45 min.

### 3.2. Rheological Characterization of the Paste

To obtain a processable paste for 3D screen printing, several prerequisites should be met by the paste. One is the shear thinning behavior, meaning that under shear stress (such as the squeegee movement in the flooding or the extrusion through the mesh in the printing process), the paste should become liquid enough to fill the mesh and be pressed through it without excessive force. However, the paste needs to regain its initial gel strength immediately after printing to obtain precise structures with defined edges (low thixotropic behavior). And the paste needs to exhibit a certain elasticity (1 to 10%) to obtain homogenous flooding without the shrinkage or tearing of the paste.

Strong shear thinning can be observed in the rotational measurements of shear viscosity depending on the shear stress of both pastes (Figure 2A). The imposed shear stress leads to the disentanglement of the macromolecules in the paste (e.g., hydroxypropyl cellulose). By employing a power law fit, a shear thinning parameter *n* of 0.74 can be determined using $\eta =K\;\cdot {\dot{\gamma }}^{n-1}$, with *η* being the viscosity, $\dot{\gamma }$ the shear rate, and *K* the consistency index (*K* = 1.07). This shear thinning behavior is crucial for processing the paste during 3D screen printing. Here, the paste needs to be liquid enough to be extruded through the mesh. After the removal of the shear force, that is when the paste is deposited on the previous layer, it should regain its strength immediately to build up a precise 3D structure. This can be simulated by a 3-step shear rate test, where a low shear rate of 0.1 s^−1^ is applied for 1.5 min and a higher shear rate of 100 s^−1^ for 0.5 min where the shear stress causes the viscosity of the pastes to drop (Figure 2B). The recovery of its viscosity, and thereby dimensional stability after the removal of the shear force, can be observed when the shear rate is again lowered to 0.1 s^−1^. With increasing shear rate, a sudden drop in viscosity can be observed; however, a steady state is not fully reached within the 0.5 min. This can be explained by the crospovidone, which is used as a disintegrant. The network of highly crosslinked polyvinylpolypyrrolidone needs time to equilibrate to the shear stress. When the shear stress is relieved, the viscosity converts back to values of >10 Pa s rather quickly, though it does not reach its initial viscosity form before the shear. This might be attributed to alterations in the microstructure induced by the shear force, e.g., the breakage of entanglements. This fast recovery is characteristic of a sample exhibiting low thixotropy. This thixotropic behavior, meaning the slower reaction to changes in shear stress, can further be observed by a shear rate ramp test from low to high shear rates and vice versa (Figure 2C). Here, the two curves are not congruent. By measuring from high to low shear rates in the second run, the obtained viscosities are lower, implying a slightly thixotropic behavior. To further elucidate the microstructure of the sample, oscillatory shear experiments have been conducted. In amplitude sweep tests, the shear modulus of the sample is measured depending on the applied strain (Figure 2D). The viscous (loss modulus G″) and elastic (storage modulus G′) components of the shear modulus give insight into the nature of the sample. If G′ > G″, the sample is of an elastic nature. If G″ > G′, the sample is viscously dominated. Up to strains of 1.25%, the elastic component G′ dominates and therefore the paste behaves like an elastic solid. Increasing strains lead to an increase in G″ and a resulting viscous behavior (G″ > G′). The length of the linear viscoelastic region (LVER) where G′ > G″ provides information on how long the paste can be elastically deformed before the onset of structural breakdown at G’ = G″ and, thus, is an indicator for the stability of the paste, e.g., resistance against sedimentation, which is an indicator for extended shelf life. Frequency measurements reveal the characteristics of a viscoelastic solid at low frequencies < 10 Hz (Figure 2E). At these frequencies, the shear modulus is nearly independent of the frequency and, thus, the paste could be classified as a well-structured gelled system consisting of a network of associated particles. These gel-like properties, which are quickly reattained after shear stress is applied, are beneficial for a good height build-up and complete demolding from the screen in the printing process. Due to the fast recovery of the paste’s stability, even a colored paste (here the blue cross) can be printed on top without the bleeding of the colors. In conclusion, the rheological properties of a shear-thinning, slightly thixotropic paste renders it suitable for the fabrication of precise patterned tablets in 3D screen printing.

### 3.3. Physical Properties of Printed Tablets

Because of the distinct characteristics of layer-by-layer additive manufacturing through 3DSP, wherein each layer can possess unique geometry and material, it becomes feasible to print any geometry or ingredients within a tablet, resulting in a product that is virtually impossible to counterfeit. To achieve the integration of a colored cross into the body of the tablet, resulting in distinct patterns upon breakage, two pastes and two screens were employed in total. This process involved two interruptions to exchange pastes and screens. Initially, 25 layers of Paste 1 were 3D screen-printed using Screen 1 to accommodate the round tablet geometry. Subsequently, a blue-colored Paste 2 was utilized to 3D print the cross pattern with Screen 2 for 5 layers. Following a switch to Paste 1 and screen 1, an additional 25 layers were printed (Figure 3). Notably, a counter-screen was not necessary to fill the space next to the cross owing to the limited dimensions of the cross and the characteristics of Paste 1.

Figure 4 demonstrates the unique features of the tablets. From the outside the identifier (blue cross) cannot be seen (Figure 4A, left). However, breaking the tablets either lengthwise (Figure 4B) or slightly off-center (Figure 4C) revealed defined patterns, which establish them as legitimate tablets. However, a thin line can be seen on the side of the tablet, corresponding to the layer directly on top of the blue cross. Here, due to the “empty spaces” between the arms of the cross, the drying and, thus, the density of the next few layers might have been slightly altered. To prevent this, a negative screen could be utilized.

In Table 4, the physical parameters of the tablets are summarized. The data revealed that the tablets are uniform in mass and in dimensions, with limited deviations for the mass (2.3%), for the diameter (0.3%) and for the height (2.9%). These data are comparable with other 3D screen printed tablets and provide evidence for the high precision of 3D screen printing [40,41,42,43].

In Table 5, the results of the galenic tablet tests are summarized. Friability was low with a 0.155% mass loss. A value of 53 N for the hardness and a tensile strength of 1.33 MPa were obtained. The tensile strength was calculated according to Fell and Newton [44]. The values are in good agreement with the ones from tablets generated with other 3D printing methods [41,45,46]. The disintegration time of 9 min corresponds well to an immediate release characteristic.

In vitro dissolution testing of the disc-shaped tablet with embedded cross (Tab01) showed the desired immediate release kinetics (Figure 5).

## 4. Conclusions

In conclusion, it could be shown that 3DSP allows for the manufacturing of tablets with a distinct inner architecture, namely the integration of a colored cross at a large scale. This feature offers anti-counterfeiting properties. In a continuous process with only 2 stops change the screen and paste, several thousand tablets were manufactured. However, if scaled up, 3DSP can produce up to 200,000 tablets per hour depending on the type of screen and machine (e.g., inline, two printing stations) used.

As a proof-of-concept, a colored paste was used for the embedded cross. Interestingly, due to the high flexibility regarding processable materials, one can contemplate the use of materials with distinct features such as micro- or nanoparticles, [26,47] DNA-strands [48], or ultraviolet inks [49]. Due to the gentleness of the process, even delicate materials are processable. The drying temperature can also be tuned to lower values at the expense of the cycle time. The combination of several anti-counterfeiting measures would tremendously enhance the safety and protection of the dosage form. Here, 3DSP offers the possibility of a combination of covert (such as the embedded cross presented herein) and overt (e.g., a QR-code) [39] features in a single process with minor intermediate steps.

Even changes in an established anti-counterfeiting feature, e.g., the embedded cross, could easily be applied by simply changing the geometry of the second screen to, for instance, a star or line pattern, or by changing the type of paste (e.g., added security inks [50,51]) to complicate the counterfeiting of the dosage form. There are no limits to the layout of the embedded feature, since very fine patterns can also be resolved using 3DSP [39].

This innovative approach demonstrates the power of 3D screen printing to implement novel and innovative anticounterfeiting measures into the manufacturing of solid oral dosage forms. The method’s simplicity and adaptability to other formulations makes it a potential solution to enhance drug security, pushing the frontiers in the fight against counterfeit drugs in favor of patients and even the stakeholders involved in the pharmaceutical industry.

## Figures and Tables

**Figure 1 pharmaceutics-16-00368-f001:**
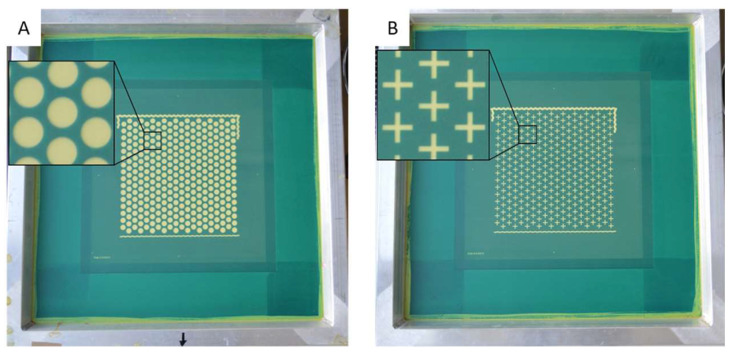
Screen layout for disk-shaped tablet geometries ((**A**), Screen 1) and cross-shaped geometries ((**B**), Screen 2). The magnifying inlets reveal the geometries of the tablet and the cross in more detail. The screen size is 584 × 584 mm^2^ with a printing area of 205 × 205 mm^2^, which can accommodate the 330 tablets at the chosen dimension. Disk-shaped tablets have a diameter of 9.14 mm.

**Figure 2 pharmaceutics-16-00368-f002:**
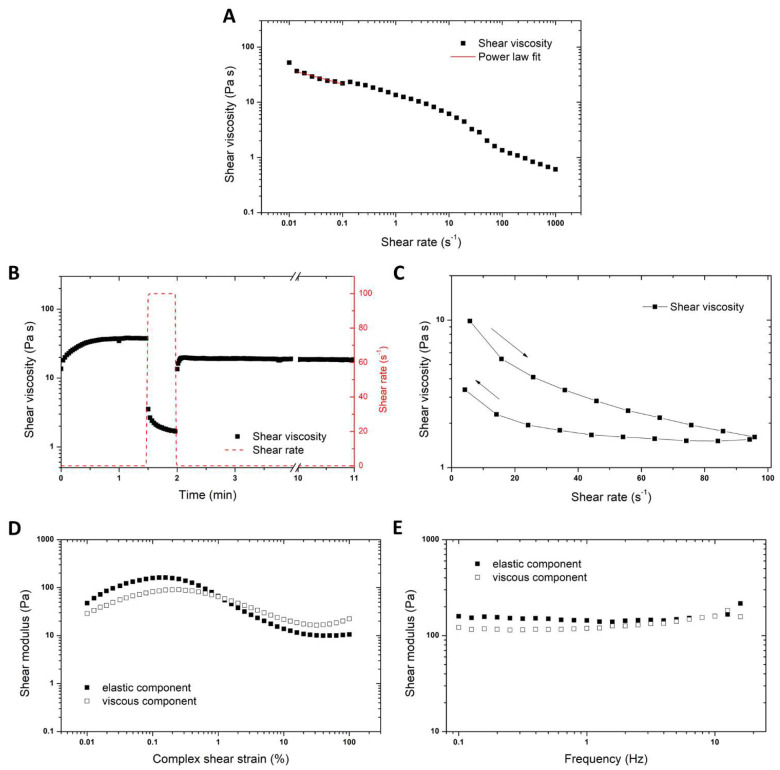
Shear stress-dependent viscosity (**A**), 3-step shear rate experiment (**B**), shear rate ramp test (**C**) (The arrows indicate the direction of measurement. Right arrow signifies measurement towards high shear rates, left arrow signifies measurement returning from high to low shear rates), amplitude sweep (frequency = 1 Hz) (**D**) and frequency sweep (strain = 0.05 %) (**E**) of the paste. The elastic and viscous components of the shear modulus are depicted as solid and hollow data points, respectively.

**Figure 3 pharmaceutics-16-00368-f003:**
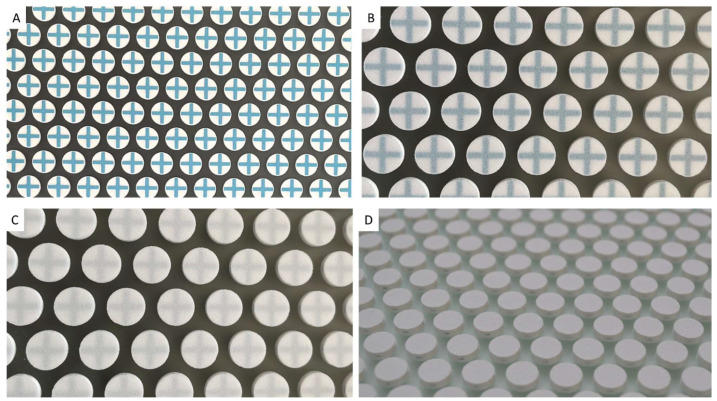
Photographs of tablets during the printing progress. Tablets with a blue cross at layer 30 (**A**), Tablets at layer 35 (**B**), Tablets at layer 40 (**C**) and tablets with a total of 55 layers after the printing process (**D**).

**Figure 4 pharmaceutics-16-00368-f004:**
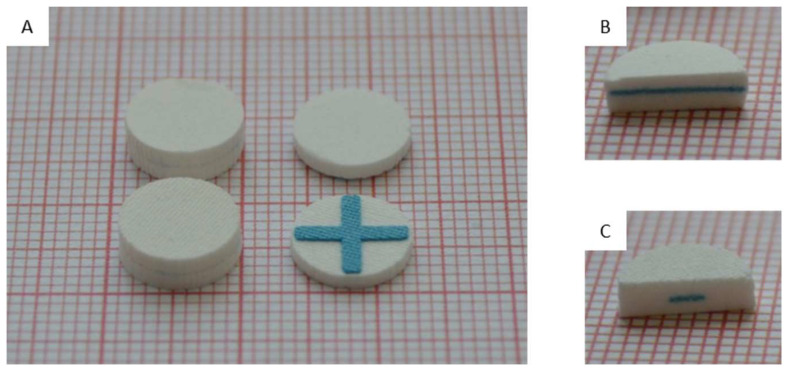
Photographs of 3D screen-printed tablets in disk-shape on millimeter paper. Finished tablet and demonstration tablet with a blue cross (**A**), tablet with an anti-counterfeiting marker broken exactly in the middle (**B**) or slightly off-center (**C**) reveals the respective parts of the blue cross.

**Figure 5 pharmaceutics-16-00368-f005:**
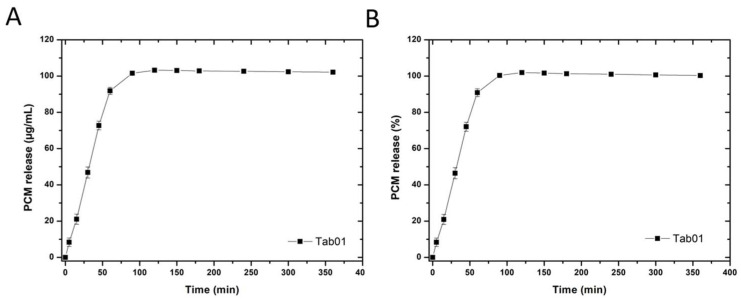
In vitro dissolution of 3D screen-printed disk-shaped tablets (Tab01) in 900 mL 0.1 M HCl, at 37 °C, 100 rpm, n = 6. Depicted is the absolute paracetamol release (**A**) and the percentage of released API over time (**B**).

**Table 1 pharmaceutics-16-00368-t001:** Composition of the immediate-release paste (Paste 1): Tablet body.

Component	Chemical Name	Function	Wt%
Paracetamol	N-(4-Hydroxyphenyl) acetamide	API	33.8
Klucel LF Pharma	Hydroxypropylcellulose	Binder	2.5
Polyplasdone Ultra 10	Crospovidone	Disintegrant	3.7
Calcium sulfate	Calcium sulfate hemihydrate	Filler, Absorber	4.2
Avicel PH-105	Microcrystalline cellulose	Filler	3.0
Pharmatose 450 M	α-d-Lactose	Filler	5.3
Glycerol	Propane-1,2,3-triol	Humectant	3.0
Talc	Magnesium silicate	Anti-tacking agent	0.3
Silfar 350	Polydimethylsiloxane	Anti-foaming agent	0.3
MilliQ Water	Water	Solvent	43.9
Total			100.0

**Table 2 pharmaceutics-16-00368-t002:** Composition of the immediate-release paste (Paste 2): Blue cross.

Component	Chemical Name	Function	Wt%
Paracetamol	N-(4-Hydroxyphenyl) acetamide	API	32.4
Klucel LF Pharma	Hydroxypropylcellulose	Binder	2.4
Polyplasdone Ultra 10	Crospovidone	Disintegrant	3.5
Calcium sulfate	Calcium sulfate hemihydrate	Filler, Absorber	4.1
Avicel PH-105	Microcrystalline cellulose	Filler	2.8
Pharmatose 450M	α-d-Lactose	Filler	5.1
Glycerol	Propane-1,2,3-triol	Humectant	2.8
Talc	Magnesium silicate	Anti-tacking agent	0.3
Silfar 350	Polydimethylsiloxane	Anti-foaming agent	0.3
TopMill Blue	Silicate, Titanium dioxide, Iron oxide	Colorant	4.2
MilliQ Water	Water	Solvent	42.1
Total			100.0

**Table 3 pharmaceutics-16-00368-t003:** Designed lateral tablet dimensions on the screen layout of screen 1 (disk-shaped tablets) and screen 2 (cross-shaped geometry) (diameter *D*, length *L*, width *W*).

Shape	*D* [mm]	*L* [mm]	*W* [mm]
Disk	9.165	-	-
Cross	-	8.50	2

**Table 4 pharmaceutics-16-00368-t004:** Physical parameters of printed disk-shaped tablets: diameter *D*, length *L*, height *H*, surface area *SA*, volume *V*, surface-area-to-volume-ratio *SA/V*, mass *m* and density *ρ*. Twenty tablets of each geometry and size were weighed and measured with a digital caliper. *SA*, *V*, *SA/V,* and *ρ* were calculated by applying the relevant formulae. The measured values are given as an average with standard deviation.

*D* [mm]	*H* [mm]	*SA* [mm^2^]	*V* [mm^3^]	*SA*/*V* [mm^–1^]	*m* [mg]	*ρ* [mg mm^–3^]
9.14 ± 0.03	2.77 ± 0.08	210.82	181.65	1.16	161.82 ± 3.67	0.89

**Table 5 pharmaceutics-16-00368-t005:** Results from the friability, hardness, and disintegration testing of disk-shaped tablets.

Tablets	Friability [%]	Hardness [N]	Tensile Strength [MPa]	Disintegration [mm:ss]
Disk with cross	0.155	52.84 ± 3.75	1.33	09:07 ± 01:36

## Data Availability

The data presented in this study are available in this article.

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
