# Peer review of "3D Screen Printing Offers Unprecedented Anticounterfeiting Strategies for Oral Solid Dosage Forms Feasible for Large Scale Production"

_pharmaceutics, 2024, doi:10.3390/pharmaceutics16030368_

Round 1

Reviewer 1 Report

Comments and Suggestions for Authors

Nicolle Schwarz et.la., presented 3D screen printing offers unprecedented anticounter-feiting strategies for oral solid dosage forms feasible for large scale production that fit well with the scope of the journal MDPI Pharmaceutics. The authors have pointed very sensitive fact, which needs to be addressed in order to stop or reduce fraud market globally.

Authors presented the manuscript in well deigned manner, However the information of usage of CDD to design the tablet structure not presented briefly, thus it is suggested to briefly include this section.

While the fabricated tablet have been quantified for several evaluations however Shape fidelity factors was missed, that need to be addressed, once authors will include the shape factors from design and export of .stl to printing technology the shape fidelity needs to verified. Thus suggested to present the result for same.

Author Response

Dear Reviewer,

Thank you very much for your valuable input. The 3D screen printing technology relies solely on structure formation due to printing of paste through a screen, whose layout is defined by open mesh areas and areas that are covered by an emulsion. Through the open mesh areas paste is extruded and the resulting pattern is only generated by the layout of the screen without the aid of additional computational calculations such as CDD. The resulting shape fidelity in this case can be correlated to the initial size of the screen openings (Table 3) and the resulting uniformity of dosage units presented in Table 4, since it can not be drawn from an .stl file.

By incooperating a short explanation in chapter 3.1 we hope we could clarify the shape generation. Please see the attachement. 

Thank you very much for all these legitimate and helpful comments. We hope we could improve our manuscript accordingly.

Reviewer 2 Report

Comments and Suggestions for Authors

The present manuscript reports on fabrication of anti-counterfeit oral dosage forms using 3D screen printing. The idea and data presented are interesting, addressing the urgent issue of counterfeit drug products.

Few points for improvement or clarification are listed below.

- Line 55. “However, FDA supports a wide range of types and authenticity“ is not fully clear. Please rephrase.

- Line 62. What would “This technique” refer to?

- Line 66. R symbols should be at the superscipt.

- Line 72. Please rephrase ”even in combination of overt and covert nature”.

- Lines 95-97. Pk changes may not only be due to changes in composition, nor necessarily result from the latter. The Authors could mention, in place of this concept or at least before it, that no alteration in the biopharmaceutical performance would be expected.

- Line 103. Klucel is a trademark.

- Line 104. “was” should be changed to “were”.

- Table 1 and 2. Because the qualitative composition of the two pastes in use only differed for the food coloring being added to paste 2, I suggest the tables be merged into a single one.

- Table 2. In the Introduction, line 95, it is stated that no changes in the API content was introduced by the addition of food coloring, which would not be supported by the percentage composition of the food coloring-containing paste shown in this table. Please check and correct as appropriate.

- Line 160. Unneeded sentence.

- Line 164. Unneeded sentence.

- The dimensions of the printed dosage forms as given in Table 3 are somewhat confusing. Would the length of the inner cross exceed the diameter of the entire unit (10 vs. 9.4 mm)?

- Figure 2 is presented before being cited in the text.

- Lines 206-250. This whole section should come after requirements in terms of viscosity of the paste have been set. If and how the paste formulations were developed to fulfill such requirements should also be discussed.

- Lines 264-265. The production rate has already been mentioned before.

- Figure 4. A line between the former and the latter printed 55 layers is evident in Figure4A. Could you please comment on that? Also, from the dosage forms laid out on graph paper, the width of the anti-counterfeit cross does not seem to correspond to 4 mm.

- Lines 284-285. “Ten tablets of each geometry and size were weighted and measured with a digital caliper” contrasts with information provided in the Methods section. Was it n=10 or n= 20? Please check and correct as appropriate.

- Line 303. Please check if this paragraph is to be formatted under a numbered list.

- Lines 306-307. When stating “Due to the gentleness of the process even delicate materials are processable”, it should not be disregarded that a rather elevated drying temperature was here employed.

- Lines 318-323. The last paragraph would benefit from revision and rephrasing.

Author Response

Dear Reviewer,

thank you very much for your valuable remarks. We changed the text in accordance to your adaptation proposals.

Since the wt% of each individual component of Paste 1 differs slightly from Paste 2 due to the additional food coloring, the tables were still separated. However, we added a sentence to clarify this issue.

Thank you very much for the remark concerning the tablet dimensions. These were corrected.

Please see the attached point-to-point answers for further detail.

We thank the reviewer for all valuable remarks, and we hope we could improve the manuscript accordingly.

Reviewer 3 Report

Comments and Suggestions for Authors

Interesting manuscript with a novel concept. Please address these comments: 

1. Provide currency of 200 billion in introduction

2. Provide explanation on 3D screen printing in introduction 

2. Define length and width of cross in tablet

3. Provide dissolution profiles for paracetamol tablets, and comment on whether this formulation/production method affects dissolution profile. This is to validate the last sentence made in Introduction.

Comments on the Quality of English Language

Editing is required. e.g. there should not be only one statement in a paragraph (lines 71-72, again 81-84). Some sentences could be better structured.

Author Response

Dear Reviewer,

Thank you very much for your relevant input on this work. 

We added the currency, put more emphasis on the description of the 3D Screen Printing in the introduction and added dissolution profiles for the tablet. The length and width of the cross have been checked and corrected.

The manuscript was edited concerning the quality of English language. 

We thank the reviewer for all valuable remarks, and we hope we could improve the manuscript accordingly.

Round 2

Reviewer 1 Report

Comments and Suggestions for Authors

The authors have tried well and reflected all the said suggestions and comments, which made the manuscript enhanced with improved readability. Thus I suggest for further consideration with acceptance.

Reviewer 2 Report

Comments and Suggestions for Authors

Revisions addressed my comments.